# High levels of infectiousness of asymptomatic *Leishmania (Viannia) braziliensis* infections in wild rodents highlights their importance in the epidemiology of American Tegumentary Leishmaniasis in Brazil

José Ferreira Marinho-Júnior[1], Juliana F. C. L. S. Monteiro[1], Ana Waléria Sales de Carvalho[1], Francisco Gomes de Carvalho[1], Milena de Paiva Cavalcanti[1], Jeffrey Shaw[2], Orin Courtenay[3]*, Sinval Pinto Brandão-Filho[1]*

1 Department of Immunology, Instituto Aggeu Magalhães/FIOCRUZ, Cidade Universitária, Recife, Pernambuco, Brazil, 2 Department of Parasitology, Institute of Biomedical Sciences, University of São Paulo, São Paulo, São Paulo, Brazil, 3 The Zeeman Institute and School of Life Sciences, University of Warwick, Coventry, United Kingdom

* sinval.brandao@fiocruz.br (SPBF); orin.courtenay@warwick.ac.uk (OC)

## Abstract

### Background

The epidemiological significance of wildlife infections with aetiological agents causing human infectious diseases is largely determined by their infection status, contact potential with humans (via vectors for vector-borne diseases), and their infectiousness to maintain onward transmission. This study quantified these parameters in wild and synanthropic naturally infected rodent populations in an endemic region of tegumentary leishmaniasis in northeast Brazil.

### Methods

Capture-mark-recapture (CMR) of rodents was conducted over 27 months in domestic/peri domestic environs, household plantations and nearby Atlantic Forest (9,920 single trap nights). Rodent clinical samples (blood and ear tissue) were tested for infection by conventional PCR and quantitative PCR (qPCR) for *Leishmania (Viannia) braziliensis*, and xenodiagnosis to measure infectiousness to the local sand fly vector.

### Results

A total 603 individuals of 8 rodent species were (re)captured on 1,051 occasions. The most abundant species were *Nectomys squamipes* (245 individuals, 41% of the total catch), *Rattus rattus* (148, 25%), and *Necromys lasiurus* (83, 14%). All species were captured in greater relative frequencies in plantations; *R. rattus* was the only species captured in all three habitats including in and around houses. Four species, comprising 22.6% of individuals captured at least twice, were geolocated in more than one habitat type; 78.6% were

**Data Availability Statement:** All relevant data are within the manuscript and its Supporting Information files.

**Funding:** The study was funded by FACEPE (Foundation for Science and Technology of Pernambuco, www.facepe.br, grant APQ 0024.4.00/13), CAPES (Coordination of Improvement of High Education, www.gov.br/capes/pt-br, PAPB Aux. 23038.005276/2011-3) and, OC was supported by the CNPq (National Conseil of Scientific Development and Technology, www.gov.br/cnpq/pt-br, Ciências sem Fronteiras Programme [Science without Borders], grant no. 400331/2012-8). The funders had no role in study design, data collection and analysis, decision to publish, or preparation of the manuscript.

**Competing interests:** The authors have declared that no competing interests exist.

infected with *L. (V.) braziliensis*, facilitating inter-species and inter-habitat transmission. Species specific period prevalence ranged between 0%-62% being significantly higher in *N. squamipes* (54–62%) and *Hollochillus sciureus* (43–47%). Xenodiagnosis was performed on 41 occasions exposing 1,879 *Nyssomyia whitmani* sand flies to five rodent species (37 individuals). Similar mean levels of infectiousness amongst the more common rodent species were observed. Longitudinal xenodiagnosis of the *N. squamipes* population revealed a persistent level of infectiousness over 13 months follow-up, infecting a median 48% (IQR: 30.1%-64.2%) of exposed blood-fed vectors. The proportion of exposed flies infected was greater in the low compared to in the high seasonal period of vector abundance. *L. (V.) braziliensis* parasite loads in rodent blood quantified by qPCR were similar across rodent species but did not represent a reliable quantitative marker of infectiousness to sand flies. The standardised risk of rodent infection in plantations was 70.3% relative to 11.3% and 18.4% in peri domestic and forest habitats respectively. *R. rattus* was the only exception to this trend indicating greatest risk in the peri domestic environment.

## Conclusions

The results support the view that a collective assemblage of wild and synanthropic rodent species is an important wild reservoir of *L. (V.) braziliensis* in this region, with *N. squamipes* and *R. rattus* probably playing a key role in transmission within and between habitat types and rodent species. Rodents, and by implication humans, are at risk of infection in all sampled habitats, but more so in homestead plantations. These conclusions are based on one of the longest CMR study of small rodents in an American Tegumentary Leishmaniasis (ATL) foci.

## Author summary

Our previous studies on the prevalence of leishmanial infections in wild caught rodents suggested that they represent potential reservoirs of *Leishmania* (*Viannia*) *braziliensis* in northeast Brazil, supported by showing that laboratory bred rodents that were experimentally infected with *L. (V.) braziliensis* were infectious to *Lutzomyia longipalpis*. The present field study reports the abundance of rodent species in different ecotopes, the prevalence of *Leishmania* infections in a cohort of naturally infected rodent species and the transmission potential of the most abundant species by xenodiagnoses using the primary vector, *Nyssomyia whitmani*. The results incriminate the local guild of rodents as a collective wildlife reservoir in the zoonotic transmission cycle of *L. (V.) braziliensis*. The greatest infection risk to rodents, and likely also to humans, is in peri domestic plantations, where *N. squamipes*, *Ne. lasiurus* and *R. rattus*, in particular, occur at high densities; the former species supported the highest infection prevalences and were infectious for extended periods. *R. rattus* was the only species captured supporting infections in and around households in addition to the other habitats. More than 20% of rodents captured at least twice moved between habitat types. There was a qualitative but not quantitative relationship between *L. (V.) braziliensis* parasite loads in rodent blood samples and xenopositive sand flies.

## Introduction

The leishmaniases are a collection of protozoal diseases of humans and animals associated with phylogenetically diverse parasites that belong to the Leishmaniinae subfamily [1]. The genus *Leishmania* comprises of more than 20 species, transmitted between mammalian hosts by principally blood-seeking female Phlebotomine sand flies, and most involving one or more non-human animal reservoir in their zoonotic transmission cycle. Depending on the infecting *Leishmania* species, the disease in humans causes cutaneous, mucocutaneous, or visceral clinical manifestations ranging from self-limiting dermatological lesions to life-threatening visceral involvement requiring complex diagnosis and treatment regimes. Currently endemic in up to 92 countries worldwide, prevention or curtailing of transmission and geographical expansion [2,3] requires a full understanding of the aetiological agents, arthropod vectors and animal hosts maintaining transmission [4].

One such challenge is the complex eco-epidemiology of *Leishmania* (*Viannia*) *braziliensis* that causes American Tegumentary Leishmaniasis (ATL). ATL is a human disease of skin and/ or mucosal tissues caused by a number of *Leishmania* parasites throughout the Americas [5–7]. Subclinical *L.* (*V.*) *braziliensis* infections have been detected in a multitude of wild, synanthropic, and domesticated animal hosts, but their epidemiological significance are unknown. The uncertainties emanate in part from misinterpretation of host infection data which alone are insufficient to differentiate reservoir hosts (sources of *Leishmania*) from non-reservoir hosts (*Leishmania* sink hosts) as infection *per se* does not necessarily equate to their transmission potential to sand fly vectors [8–10].

Advances in our understanding are further hindered by the lack of experimental studies due to difficulties to establish *L.* (*V.*) *braziliensis* in culture [11], to maintain detectable infections in laboratory animal models [6,12], and standardisation issues associated with the diversity of genetic and molecular profiles of the parasite, coupled with the non-uniformity in the clinical behaviour in laboratory animals [13,14]. Consequently, compared to the other *Leishmania* species, there are few experimental studies of *L. (V.) braziliensis*. Moreover, in the absence of longitudinal studies of naturally infected host populations, it is not clear if the course of laboratory infections indeed mimic those in nature.

Almost nothing is known about wildlife host infectiousness to sand fly vectors, and how it relates to host parasite loads and tissue tropism, and if quantitative PCR (qPCR) methods to detect *Leishmania* genome equivalents in host tissues offers a reliable marker of host infectiousness by which to measure transmission potential at the population-level; xenodiagnosis, the standard, is difficult to perform at scale.

Here, we present the results of a longitudinal capture-mark-recapture (CMR) study of wild and synanthropic rodents naturally infected with *L. (V.) braziliensis* in northeast Brazil. The specific objectives were to (i) document rodent species abundance in different ecotypes associated with human activity; (ii) quantify their infection prevalence; (iii) measure their infectiousness to the sand fly vector; (iv) determine the relative infection risk by ecotype; (v) investigate the potential of tissue parasite loads as a surrogate marker of host infectiousness.

## Methods

### Ethical considerations

Study protocols were approved by the Ethics Committee on Animal Use (CEUA / IAM) (No. 017/2011), and permission from IBAMA (Brazilian Institute of Environment and Renewable Natural Resources) through the Authorization for Activities with Scientific Purpose (No. 12749).

## Study area

Fieldwork was undertaken in two rural endemic communities, Engenho Raiz de Dentro (8˚ 42.7944'S, 35˚ 44.7078'W) and Engenho Refrigério (8˚ 42.7963'S, 35˚ 49.4742'W) located 8.73 km apart, in Amaraji municipality in the State of Pernambuco, north-east Brazil (S1 Fig). The municipality is located south of Zona da Mata about 100 km from Recife, 385m above sea level, and belongs to the north-eastern coastal region known as the Coastal Plateaus (Tabuleiros). The municipality population is approximately 22,829 habitants of which 27% live in rural areas [15,16].

Local temperatures range from 18˚C to 29˚C, with an average annual rainfall of at least 1,000mm, and a high degree of humidity throughout the year (Köppen–Geiger climate classification: Aw—Tropical wet and dry). The municipality of Amaraji is situated in the domains of the Serinhaém River Basin fed by small streams and water sources. Originally the land was divided into extensive sugar cane plantations known as Engenhos (Mills), of which some were expropriated by the government and divided into plots (˜Parcels˜) to be occupied by local farmers who became their owners. This led to an increase in the rural population living in scattered small homesteads (S2 and S3 Figs) and attending their plantations principally of bananas, pineapples and vegetable cash crops. Pockets of sub-deciduous Atlantic forest typically occur on the steep-sided hill tops of oxisols and podzolic soils, while the alluvial soils in the valleys with small rivers and floodplains are preferred for agriculture. Residents regularly work the homestead plantations whereas the commercial sugarcane plantations only receive seasonal workers between the months of November to April for harvesting. Occasionally the plantations are sprayed with insecticides.

In northeast Brazil, the known sand fly vectors of ATL are *Nyssomyia whitmani* and *Migonemyia migonei* [17,18]. In this study site only *Ny. whitmani* is present representing >97% of the total sand flies captured monthly over two years; seasonal low abundance variably occurs during 3–5 months from May to July/October [19], which roughly corresponds to the coldest (August and July) and the driest (September to December) months.

Between 2011 and 2014, 1,082 ATL cases were reported in Pernambuco including 14 in the Amaraji municipality. *Leishmania* skin test surveys in the region have shown that the actual number of infections is substantially higher, and that the numbers of reported clinical cases does not reflect the levels of transmission [20]. The apparent reductions in case numbers in recent years is partially explained by the high prevalence of acquired immunity against clinical infection.

## Small mammal trapping regime

Between May 2012 and August 2014, 20 rounds of CMR were conducted each of two to 11 days duration (S1 Table). Most of the trapping effort (89.5%) was in Engenho Raiz de Dentro due to seasonally limited access to Engenho Refrigério. 15 homesteads located 200-500m apart, as typical in this region, were selected as central reference points for sampling, chosen due their proximity to peri domestic plantations and Atlantic forest patches in order to facilitate repeat trapping. Eighty numbered Tomahawk traps (45cm x 21cm x 21cm) were set per night, 5 traps in domestic habitats (inside houses and outside within the household boundary- labelled here as peri domestic sites), 55 in adjacent plantations, and 20 in Atlantic Forest remnants. Traps were placed in a single line 2m apart in forest and plantations, and placed in the kitchen, bedroom and/or living room inside houses, and in the yard, terrace and/or near animal shelters outside houses. Traps were placed in the same reference points on each CMR round. Traps were baited with pieces of banana and sugar cane.

## Animal processing

Captured animals were identified to species using the nomenclature following previous published references to *Akodon arviculoides*, *Oryzomys eliurus*, *Oryzomys subflavus* and *Oxymycterus angularis* and herein referred to as *Akodon cursor*, *Oligoryzomys nigripes*, *Cerradomys subflavus* and *Oxymycterus dasytrichus*, respectively [21]. Animals were sexed and assigned to an age-class at first capture (YY- juvenile; JA- young adult; AA-mature adult) based on their body size by experienced field technicians. Rodents were anesthetized with recommended doses of pre-anaesthetic Xylazine 2%, and aesthetic Ketamine 10%. A blood sample of 0.5 to 1mL was obtained from the retrobulbar venous sinus with a sterile Pasteur pipette. A skin fragment was collected from the right ear of each animal by biopsy punch to provide a 25mg skin sample. Any ectoparasites were removed. Animals were then marked with an individual nano microchip (1.25mm x 7.0mm) (Micro-ID, Burgess Hill, West Sussex, UK) read using a Micro-ID hand scanner (Model MIDO1C). Animals were then placed in micro-isolators in the field station for recovery. The next morning when fully alert the rodents were released at the site of capture. Animals that were subsequently re-captured within the same sample round were identified, recorded and released without further diagnostic sampling. Recaptured individual animals were diagnostically sampled at a median interval of 50 (IQR: 35–91) days. During the study, 32 marsupials were also captured, comprising 21 *Didelphis albiventris* (1 inside a house; 15 in plantations; 5 in forest), 6 *Marmosa* spp., (3 in plantations; 3 in forest) and 5 *Monodelphis domestica domestica* (5 in plantations). As marsupials were not the focus of the study they were not sampled and released immediately.

## Xenodiagnosis

**Xenodiagnoses using wild-caught *Ny. whitmani* sand flies.**   Xenodiagnosis was performed using wild *Ny. whitmani* sand flies which were trapped by CDC light traps (set 6pm-6am) in animal shelters, stables and chicken coops in the nearby locality of Engenho Raiz Nova where *Ny. whitmani* is the dominant sand fly species [22]. After separating the blood-fed and non-blood-fed females, the latter were placed in a Barraud cage into which anaesthetised rodent(s) were introduced and flies allowed to feed for 1 hour in darkness. The rodent(s) submitted for xenodiagnosis on any single day depended on the number of rodents and sand flies captured on the night previous. Post exposure, blood-fed and un-fed female sand flies were removed from the cage and separated individually into a tube containing 70% ethanol for subsequent DNA extraction and qPCR testing. The aim was to expose each captured rodent to a minimum of 20 female *Ny. whitmani*. In the final analysis, a median 34 (95% C.L. 19–48) blood-fed female flies were tested from each xenodiagnosis experiment.

For analyses of the xenodiagnosis data using wild-caught *Ny. whitmani*, the observed proportion of sand flies infected with *L. (V.) braziliensis* was adjusted to account for an expected background level of *Leishmania* natural infection estimated from an independent sample of 250 *Ny. whitmani* captured during contemporary trapping rounds in the same area. Of these, 6% (15/250) proved infected, hence the value of the upper 95% confidence limit (C.L.) of the mean infection rate in this control sample was subtracted from the observed fly infection rates by xenodiagnosis. Similarly, the upper 95% C.L. of the *L. (V.) braziliensis* parasite loads estimated by qPCR in the naturally infected sand flies was subtracted from the values detected in xenopositive *Ny. whitmani*.

**Xenodiagnoses using laboratory reared *Lutzomyia longipalpis*.**   For comparison with a likely secondary sand fly vector species, *Lutzomyia longipalpis* females were obtained from a laboratory colony held at IAM/FIOCRUZ established from flies captured in Passira municipality, located in the Agreste zone of Pernambuco State, and reared as described [23]. Rodents were anesthetized and placed in cages as above with *Lu. longipalpis* unfed females for 1 hour.

After removal, groups of 25 blood-fed female sand flies were maintained in separate Barraud cages supplied with sugar solution for five to 7 days to allow parasite development. They were then stored individually in 70% ethanol for subsequent DNA extraction and qPCR testing. A mean 81 (95% C.L. 42, 121) *Lu. longipalpis* females were tested from each xenodiagnosis experiment. As *Lu. longipalpis* sand flies were laboratory bred, adjustments to infection rates as above were not required.

## Molecular tests

DNA was extracted from 200μL blood samples using QIAamp DNA Mini Kit Blood (QIAGEN) Ⓡ and from 25mg skin biopsy samples using QIAamp DNA Mini Kit (QIAGEN)Ⓡ. The conventional PCR was performed as described previously [24] using the B1 and B2 kDNA primers (B1: 5′-GGGGGTTGGTGTAATATAGTGG-3′ / B2: 5′-CTAATTGTGCACGGGGAGG-3′) as previously described [25]. The qPCR protocol's final volume was 50 ml containing primers [kDNAf-ATGCCTCTGGGTAGGGGCGTTC/kDNAr1-GGGAGCGCGGCCCACTATATT] (5 pmol/ml) and 2X SYBRGreen Master Mix (Applied Biosystems) [26,27]. For qPCR, a standard curve was included on each plate using *L. (V.) braziliensis* reference strain LB2903 at seven serial dilutions (1ng, 100pg, 10pg, 1pg, 100fg, 10fg, to 1fg). The test sample OD values were compared to the standard curve and converted into arbitrary units as reported hereafter. Ct values <33 were considered valid based on prior validation of qPCR on each sample type. Test samples were run in duplicate using standard conditions (40 cycles of denaturation at 95˚C for 15s and annealing at 60˚C for 1min), in ABI Prism 7500 Sequence Detection System (Applied Biosystems) with 7500 Software (version 2.0.5) [27].

## Statistical analyses

Estimates of infection and infectious status were computed by fitting (0/1) data to mixed-effects binomial complimentary log–log models including rodent ID and time (days) from first capture as random effects, and reported as risk ratios (RR). In cases where models failed to converge, a cluster term (animal ID) was included to control for likely autocorrelation. Models included variables characterising time (days) since first capture, month, and year of trapping (3 levels), age-class (3 levels), sex (2 levels), ecotype (3 levels) and rodent species (8 levels), as appropriate. Similarly, the proportion of sand flies infected by xenodiagnosis was modelled as binomial data using Generalise Linear Models.

Complimentary log–log model fits were achieved by Gauss–Hermite numerical adaptive quadrature of the random-effects estimators (quadchk routine in STATA), validated using 16 integration points in model run comparisons to confirm quadrature fitting accuracy; model runs showed 0.01% variation in resulting estimates, and were thus considered to be reliable. Potential explanatory variables were evaluated in a preliminary step by log–likelihood ratio test (LRT) of nested models.

Data were analysed in STATA v.15 (StataCorp LP, College Station, TX).

## Infection risk

To standardise the infection risk relative to trapping effort and trapping success between ecotypes where captured, crude capture rates were adjusted by:

$$\frac{a_{ij}}{\sum_{j=1}^{3} a_{ij}} \cdot \frac{p_{ij}}{n_{ij}} \tag{1}$$

where $a_{ij} = \frac{a}{b}$ given

$a$ = ∑ number of successful capture nights of rodent species $i$ within ecotype $j$

$b$ = ∑ number of attempted trap nights within ecotype $j$

$i$ = the rodent species

$j$ = the ecotype

$p$ = the number of rodent capture events that showed evidence of infection

$n$ = ∑ number of rodent capture events where clinical samples were diagnostically tested

## Results

### Sampling

Between May 2012 and August 2014, a total 9,920 single trap nights were accomplished in 20 trapping rounds over 27 months. Of these 620 trap nights were in the peri domestic environment, 6,820 in plantations and 2,480 in forest patches, a ratio of 1: 11: 4 respectively (Tables 1 and S1).

### Trapping success

A total 603 rodent individuals of eight species were (re)captured on 1,051 occasions, of which the most abundant species, indicated by the capture records, were *Nectomys squamipes* (n = 245 individuals representing 41% of captured individuals), followed by *Rattus rattus* (n = 148, 25%), and *Necromys lasiurus* (n = 83, 14%) (Table 1).

New individuals were captured and recaptured over much of the trapping period, with a noticeable fall-off in trapping success in the latter four trap rounds in part associated with the reduced trapping effort (Fig 1 and S1 Table). With exception of *Ol. nigripes* and *C. subflavus* for which only 2–4 individuals were captured (none recaptured), the median number of recapture events per individual rodent was generally similar across species, as was the median recapture interval (45 to 69 days) (S2 Table). *N. squamipes* was the exception and most common species recaptured up to 11 times each in plantations and forest habitats (S4A and S4C Fig),

**Table 1. Rodent trapping success: number and proportion of total (re)capture events across sampled ecotypes.**

| Rodent species | Number (proportion[1]) of total capture events across habitats | | | Total individuals captured (proportion of total individual rodents captured) | Total capture events (proportion of total capture events) |
|---|---|---|---|---|---|
| | Peridomestic | Plantation | Forest | | |
| *A. cursor* | 0 (0.00) | 34 (0.71) | 14 (0.29) | 32 (0.053) | 48 (0.046) |
| *H. sciureus* | 0 (0.00) | 62 (1.00) | 0 (0.00) | 40 (0.066) | 62 (0.059) |
| *Ne. lasiurus* | 0 (0.00) | 112 (0.87) | 17 (0.13) | 83 (0.138) | 129 (0.123) |
| *N. squamipes* | 0 (0.00) | 481 (0.88) | 63 (0.12) | 245 (0.406) | 544 (0.518) |
| *Ol. Nigripes* | 0 (0.00) | 1 (0.50) | 1 (0.50) | 2 (0.003) | 2 (0.002) |
| *C. subflavus* | 0 (0.00) | 3 (0.75) | 1 (0.25) | 4 (0.007) | 4 (0.004) |
| *O. dasytrichus* | 0 (0.00) | 67 (0.91) | 7 (0.09) | 49 (0.081) | 74 (0.070) |
| *R. rattus* | 56 (0.30) | 111 (0.59) | 21 (0.11) | 148 (0.245) | 188 (0.179) |
| **All species** | 0.05 | 0.83 | 0.12 | | |
| **total (re)capture events** | 56 | 871 | 124 | 603 (1.00) | 1,051 (1.00) |
| **trapping success (N single trap nights)[2]** | 0.090 (620) | 0.128 (6,820) | 0.050 (2,480) | 0.061 (9,920) | 0.106 (9,920) |
| **Adjusted trapping success[3]** | 0.337 | 0.476 | 0.187 | | |

[1] the proportion of captures within the ecotype of the total (re)captures for an individual rodent species

[2] calculated as the total number of successful capture events of all species / total single trap nights within the ecotype

[3] calculated as a proportion of total trapping success across ecotypes accounting for trapping frequency and distributions of capture events within and across ecotypes.

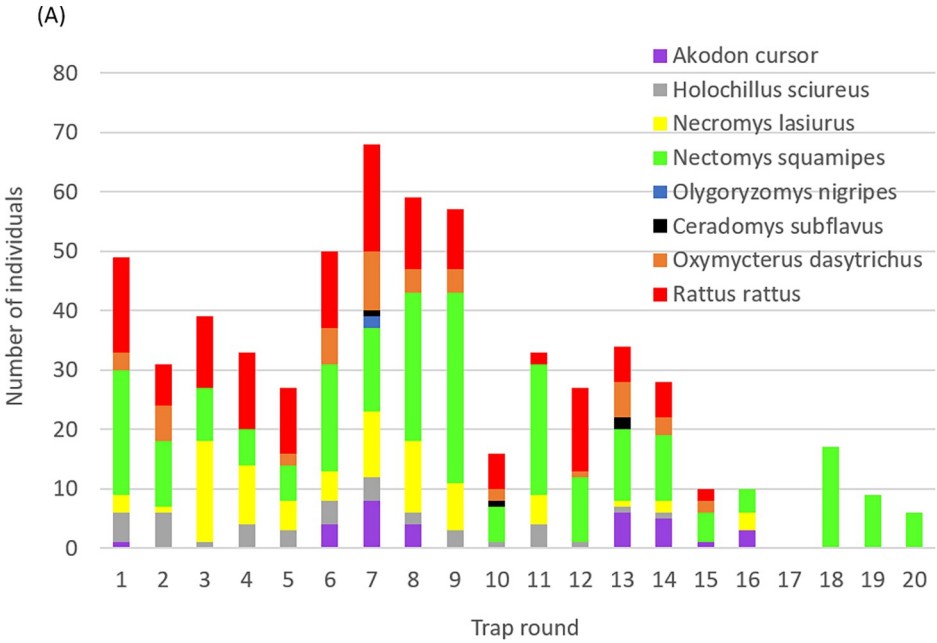

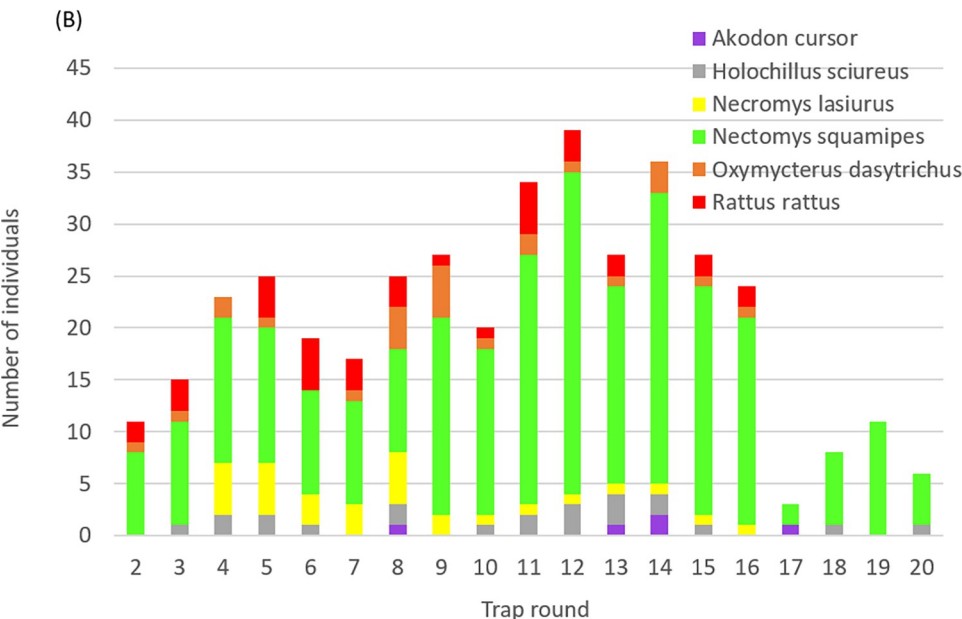

**Fig 1. Number of rodents captured in each trapping round (A) individuals at first capture; (B) all individuals recaptured.**

which was more frequent than the other species, each being recaptured up to 4 times (S4A and S4B Fig). *R. rattus* was recaptured on relatively fewer occasions and showed a greater affinity to peri domestic and plantation environments (S4D Fig), and was the only species captured in all three sampled habitat types (Table 1). The abundance and affinity with a particular habitat varied between rodent species, however the proportion of total (re)capture events was generally highest in plantations when adjusted for trapping effort (Table 1 and S4C and S4D Fig).

**Table 2. Number and proportion of rodents and clinical samples classified as infected[1] and positive by qPCR for *L. (V.) braziliensis*.**

| Rodent species | Number classified as infected / total tested (proportion)[1] | | Number of clinical samples positive by qPCR/ tested (proportion) | |
| --- | --- | --- | --- | --- |
| | animals | samples | blood | skin |
| *A. cursor* | 4/32 (0.125) | 5/37 (0.135) | 5/34 (0.147) | 0/36 (0.0) |
| *H. sciureus* | 17/40 (0.425) | 29/62 (0.468) | 21/60 (0.350) | 2/51 (0.039) |
| *Ne. lasiurus* | 28/83 (0.337) | 34/113 (0.301) | 34/112 (0.304) | 0/89 (0.0) |
| *N. squamipes* | 150/244 (0.615) | 283/523 (0.541) | 254/513 (0.495) | 11/401 (0.027) |
| *Ol. nigripes* | 0/2 (0.0) | 0/2 (0.0) | 0/2 (0.0) | 0/2 (0.0) |
| *C. subflavus* | 0/4 (0.0) | 0/4 (0.0) | 0/4 (0.0) | 0/4 (0.0) |
| *O. dasytrichus* | 16/49 (0.327) | 21/74 (0.284) | 21/74 (0.284) | 0/55 (0.0) |
| *R. rattus* | 41/148 (0.277) | 49/184 (0.266) | 47/184 (0.255) | 1/157 (0.006) |
| **Totals** | 256/602 (0.425) | 421/999 (0.421) | 382/983 (0.389) | 14/795 (0.018) |

[1]classified as infected at any sample round if positive by one or more diagnostic test: conventional PCR of blood samples, qPCR of blood or ear skin tissue, or xenodiagnosis.

Standardised for ecotype and rodent species, trapping success was 33.7%, 47.6% and 18.7% in the peri domestic environment, plantations and forest habitats, respectively (Table 1). Trapping success was significantly higher in plantations than in the peri domestic environment ($\chi^2_1 = 7.28$, P = 0.007), and higher in peri domestic than forest habitat ($\chi^2_1 = 14.75$, P = 0.0001).

## Infection prevalence

Rodents were defined as infected at each (re)capture if positive by one or more diagnostic test including conventional PCR (blood samples), qPCR (blood or ear skin tissue samples), and xenodiagnosis. Of the 602 animals and 999 sample test results, 42.5% and 42.1% respectively were considered infected (Table 2). Omitting *Ol. nigripes* and *C. subflavus* (2 and 4 diagnostic samples only respectively), all species showed a lower period prevalence of infection compared to *N. squamipes* (z>-3.26, p<0.001), and *Holochillus sciureus* (z>-2.01, p<0.045), whereas those of *N. squamipes* and *H. sciureus* were not dissimilar (z = -0.77, p = 0.442) (Table 2).

The proportion of blood samples that tested positive by qPCR (38.9% [382/983]) was significantly greater than positive by conventional PCR (5.8% [58/999]) (Tables 2 and S3). QPCR sensitivity also differed between clinical samples: only 1.8% of skin tissue samples tested positive compared to 38.9% of blood samples (Table 2). Of the 14 animals with positive skin samples, 11 were negative by blood sample. Based on conventional PCR alone, the infection prevalence of *H. sciureus* was greater than that of all other species including *N. squamipes* (z>2.35, p<0.019) (S3 Table).

## Infection risk through time

*L. (V.) braziliensis* infection risk in *N. squamipes* significantly increased from time of first capture (z = 2.47, p = 0.014), illustrated by the rise in infection prevalence amongst captured animals at sequential trapping rounds (Fig 2A). Similarly, initial increases were observed in *R. rattus* (Fig 2A) and in *H. sciureus*, *Ne. lasiurus*, and *O. dasytrichus* (Fig 2B) (z>3.56, p<0.0001). In the case of *R. rattus*, *H. sciureus* and *Ne. lasiurus* this rise was followed by a significant decrease in infection risk by the end of the study (test of days[2] quadratic term: z>-2.62, p<0.009); no such decline was observed for *N. squamipes* or *O. dasytrichus*. The risk of rodent infection with time from first capture did not significantly vary between ecotypes for

A)

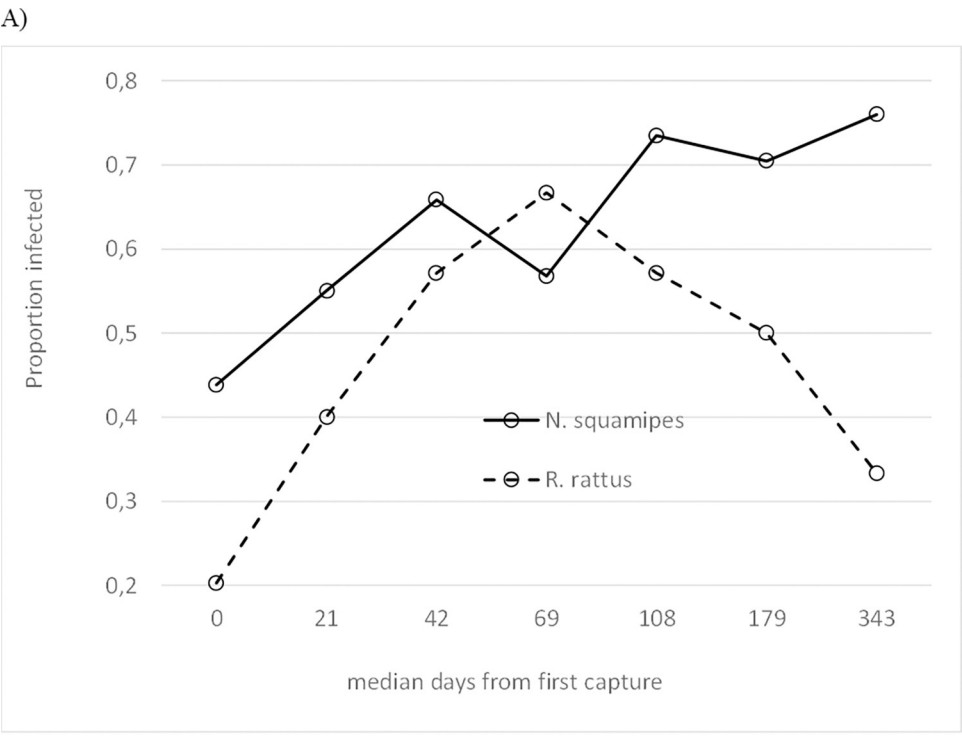

B)

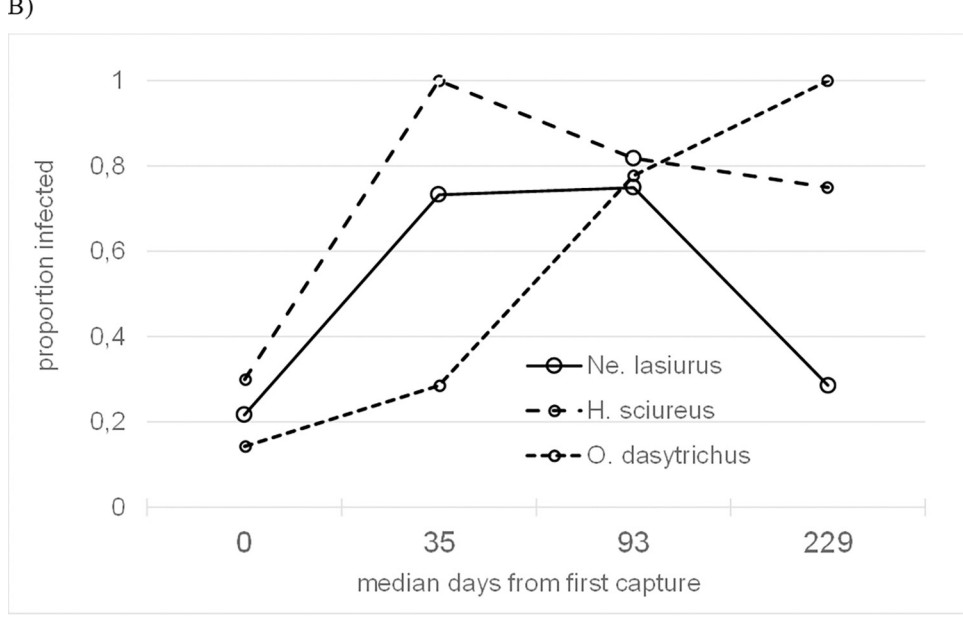

**Fig 2. The proportion of individuals infected from time of first capture.** (A) *N. squamipes* and *R. rattus*; (B) *H. sciureus*, *Ne. lasiurus*, and *O. dasytrichus*. Data were aggregated over the time frame to achieve adequate numbers.

any of the rodent species with adequate data (test of ecotype × days interaction term: z<1.36, p>0.175 in each case).

Seasonal variation in infection prevalence by calendar month was indicated for *N. squamipes* and for the other species collectively, being highest in April to July, and again in

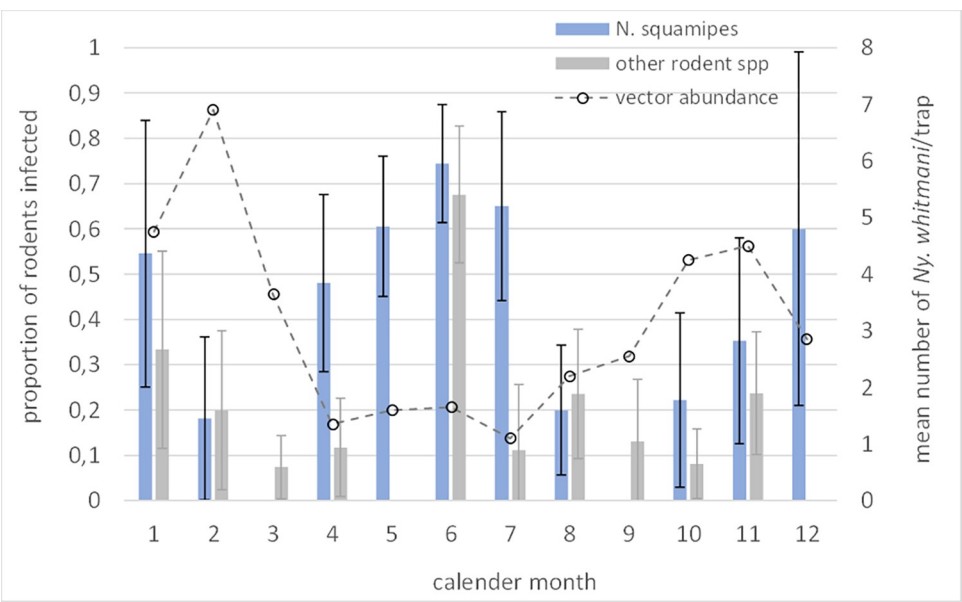

**Fig 3. Proportion of rodent populations infection by calendar month.** Prevalence calculated for *N. squamipes*, and other rodent species by aggregating data. Superimposed is the mean number of *Ny. whitmani* sand flies per month collected by CDC light trap in this and nearby foci (published data from Brandão-Filho et al 1994 [19]).

December-January (Fig 3). This pattern did not correspond to the previously reported seasonal variation in *Ny. whitmani* abundance [28], as also shown in Fig 3.

## L. (V.) braziliensis parasite loads in rodent clinical samples

*L. (V.) braziliensis* parasite loads in infected clinical samples were highly variable and the frequency distributions over-dispersed (Table 3 and S5A and S5B Fig). There was poor correlation between $\log_{10}$ transformed parasite loads in skin and blood samples from the same animal ($r_s^2 = 0.087$, p = 0.575). Median parasite loads in blood samples tended to be lower in *H. sciureus*, *N. squamipes*, *O. dasytrichus* and *R. rattus* compared to in *Ne. lasiurus* (S6 Fig), though the differences failed to reach statistical significance (z<1.95, p>0.051). There were no differences in blood parasite loads between any of the other species (z<-1.12, p>0.261). Considering the rodent populations as a whole, there was a rise in mean $\log_{10}$ transformed parasite loads in blood samples from time of 1st observed infection (z = 5.35, p<0.000), followed by a significant decline (z = -4.52, p<0.001).

**Table 3. Geometric number of L. (V). braziliensis parasites in rodent clinical samples estimated by qPCR.**

| | Geometric mean parasites per unit (95% C.I.)[2] | |
|---|---|---|
| **Rodent species[1]** | Blood (per 0.2μL) | Skin (per 25mg) |
| *A. cursor* | 11 (1.4, 79.4) | |
| *H. sciureus* | 23 (6.9, 76.5) | 12 (<0.1, >252,000) |
| *Ne. lasiurus* | 97 (38.1, 248.9) | |
| *N. squamipes* | 38 (27.5, 52.3) | 123 (8.4, 1779.6) |
| *O. dasytrichus* | 18 (5.7, 57.6) | |
| *R. rattus* | 28 (14.3, 54.9) | 26 (-) |

[1]None of the samples from *Ol. nigripes* or *C. subflavus* tested qPCR positive.
[2]Sample sizes are shown in Table 2.

**Table 4. Number (n) and proportion of (re)capture events classified as infected[1] by rodent species and ecotype.**

| Species | Peri domestic | Plantations | Forest | Total |
|---|---|---|---|---|
| *A. cursor* | 0 (0) | 0.087 (23) | 0.214 (14) | 0.135 (37) |
| *H. sciureus* | 0 (0) | 0.468 (62) | 0 (0) | 0.468 (62) |
| *Ne. lasiurus* | 0 (0) | 0.323 (96) | 0.176 (17) | 0.301 (113) |
| *N. squamipes* | 0 (0) | 0.537 (460) | 0.571 (63) | 0.541 (523) |
| *Ol. nigripes* | 0 (0) | 0 (1) | 0 (1) | 0 (2) |
| *C. subflavus* | 0 (0) | 0 (3) | 0 (1) | 0 (4) |
| *O. dasytrichus* | 0 (0) | 0.299 (67) | 0.143 (7) | 0.284 (74) |
| *R. rattus* | 0.286 (56) | 0.262 (107) | 0.238 (21) | 0.266 (184) |

[1]Infection defined as positive by one or more diagnostic method: PCR, qPCR or xenodiagnosis

## Infection and demography

Controlling for study design variables and rodent species and individual IDs, the generalised probability of infection across rodent species was not dissimilar between sexes (males: 101/327 *vs* females: 75/275; z = 0.62, p = 0.533), nor between age-classes categorised at first sample: juvenile (JJ) 47.1% (72/153); young adult (JA) 48.8% (137/281); mature adult (AA) 37.5% (212/565) (z<1.74, p>0.082). For individual species with adequate samples, there were no differences between age-classes or sex in *N. squamipes*, *R. rattus* (z<1.87, p>0.06), or *H. sciureus* (Pearson's $\chi^2_1$<0.80, p>0.36), whereas the risk of infection in *Ne. lasiurus* and *O. dasytrichus* at first sample were similarly greater in both young adult and mature adult age-classes compared to in juveniles (z>10.3, p<0.0001).

## Infection by ecotope

The three ecotopes appeared to support similar levels of infection (Table 4). However, the period prevalence varied between rodent species and between ecotopes: whereas *R. rattus* showed a similar infection prevalence across the three ecotopes, potentially higher prevalences were seen in *A. cursor* in forest compared to in plantations, and the opposite for *O. dasytrichus* and *Ne. lasiurus*, though no statistical differences were detected (Table 4).

Applying Eq 1 to account for capture success and trapping effort, a standardised proportional infection risk was calculated for each rodent species by ecotope (Table 5). Plantations represented the highest percentage (70.3%) of the total estimated infection risk compared to 11.3% in the peri domestic environment and 18.4% in forests. For individual species,

**Table 5. Proportional risk[1] of infection in rodent species in sampled habitats.**

| Rodent species | Peri domestic | Plantations | Forest |
|---|---|---|---|
| *A. cursor* | 0 | 0.041 | 0.114 |
| *H. sciureus* | 0 | 0.468 | 0 |
| *Ne. lasiurus* | 0 | 0.228 | 0.052 |
| *N. squamipes* | 0 | 0.395 | 0.151 |
| *Ol. Nigripes* | 0 | 0 | 0 |
| *C. subflavus* | 0 | 0 | 0 |
| *O. dasytrichus* | 0 | 0.232 | 0.032 |
| *R. rattus* | 0.224 | 0.037 | 0.018 |
| **Sum (% of total risk)** | 0.224 (11.3) | 1.401 (70.3) | 0.366 (18.4) |

[1]Calculated adjusted for trapping effort and trapping success by Eq 1.

**Table 6. Number (proportion) of individual rodents captured in more than one ecotype.**

| Rodent species | Number of individuals (proportion)[1] | Domestic | plantation | forest | Number animals infected |
|---|---|---|---|---|---|
| *A. cursor* | 1/3 (0.333) | | P | P | 1/1 |
| *H. sciureus* | 0/13 (0) | | | | |
| *Ne. lasiurus* | 1/20 (0.050) | | P | P | 0/1 |
| *N. squamipes* | 21/106 (0.198) | | P | P | 21/21[a] |
| *Ol. nigripes* | 0/0 | | | | |
| *C. subflavus* | 0/0 | | | | |
| *O. dasytrichus* | 0/14 (0) | | | | |
| *R. rattus* | 15/30 (0.500) | P | P | | 9/15[a] |
| *R. rattus* | 2/30 (0.067) | P | | P | 2/2 |
| *R. rattus* | 2/30 (0.067) | | P | P | 0/2 |
| total *R. rattus* | 19/30 (0.633) | | | | 11/19 |
| all species | 42/186 (0.226) | | | | 33/42 (0.786) |

P present

[1] proportion of total individuals captured at least twice.

[a] the two individuals tested by xenodiagnosis were shown to be infectious.

plantations represented a higher risk of infection than in forests; *R. rattus* was the only species that differed from this pattern (Table 5).

## Ecotype loyalty

Of the 186 individuals captured on at least two independent trap rounds, 42 (22.6%) were captured in more than one ecotype, including 21/106 (19.8%) of *N. squamipes* and 19/30 (63.3%) of *R. rattus* (Table 6). Of these, 33/42 (78.6%) were classified as infected, and at least 2 *N. squamipes* and 2 *R. rattus* tested were xenopositive.

## Infectiousness to sand flies

Xenodiagnosis was performed 44 times by exposing 5 rodent species to wild-caught *Ny. whitmani* sand flies, targeting *N. squamipes* for repeat sampling (n = 39) over an approximate 13-month period (Table 7). Thirty-nine of the 44 xenodiagnosis trials were conducted on

**Table 7. Infectiousness of free-ranging wild rodents to wild caught *Ny. whitmani* sand flies, or to laboratory-bred *Lu. longipalpis*, measured by xenodiagnosis.**

| Sand fly / Rodent species | Number of animals xenopositive / tested | Number of xenodiagnostic trials positive/tested | Proportion sand flies positive (number positive / exposed flies)[1] |
|---|---|---|---|
| *Nyssomyia whitmani* | | | |
| *A. cursor* | 1/1 | 1/1 | 0.66 (6/9) |
| *Ne. lasiurus* | 1/1 | 1/1 | 0.45 (11/25) |
| *N. squamipes* | 25/27 | 36/39 | 0.50 (686/1361) |
| *O. dasytrichus* | 1/1 | 1/1 | 0.01 (1/45) |
| *R. rattus* | 2/2 | 2/2 | 0.64 (20/32) |
| *Lutzomyia longipalpis* | | | |
| *N. squamipes* | 3/3 | 3/3 | 0.33 (72/220) |
| *R. rattus* | 2/2 | 2/2 | 0.49 (92/187) |
| Total xenopositive / tested (proportion) | 35/37 (0.946) | 46/49 (0.939) | 888/1879 (0.473) |

[1] Data aggregated across all point xenodiagnosis trials conducted on individual animals

plantation-captured animals which excluded the possibility to test for differences in infectiousness between habitat types.

Of the 41 xenopositive trials conducted using *Ny. whitmani*, 34 trials were xenopositive at the time when the animals tested positive by qPCR/PCR, and 7 trials xenopositive (all on *N. squamipes*) when the animal tested qPCR/PCR negative. Two animals exposed when qPCR/PCR positive were xenonegative. Details for individual rodents, qPCR/PCR results and the number of sand flies tested, are available in supplementary material (S4 Table).

Five xenodiagnostic trials were conducted using colony reared *Lu. longipalpis* exposed to three *N. squamipes* and two *R. rattus* individuals, all of which were xenopositive (Tables 7 and S5). All these animals except one *N. squamipes* were PCR/qPCR positive at the time of xenodiagnosis (Tables 7 and S5).

### *N. squamipes* infectiousness to *Ny. whitmani*

Over the 13-month xenodiagnosis study period, members of the *N. squamipes* population were infectious (Fig 3). The proportion of sand flies that individuals infected varied from 0.0 to 0.657. Aggregating individual data from the time of 1st capture (n = 4–12 xenodiagnosed individuals in each of 5 periods over 388 days), neither the population mean probability of being infectious, nor the proportion of total sand flies that they infected, significantly vary with increasing time (z<-0.970, p>0.146). The median proportion infected was 0.48 (IQR: 0.301–0.642); range: 0.00–0.657, N = 39), illustrated in Fig 3). Aggregating the xenodiagnosis data according to the periods of vector seasonal abundance (low season: variably May-July/October; high season: August/November-May), the proportion of exposed flies infected was higher in the low compared to in the high season (z>-9.14, p<0.001, varying the delimiting months defining the season). The respective median proportions infected were 0.64 of 716 flies (IQR: 0.600–0.657) in 23 trials, and 0.08 of 645 flies (IQR: 0.024–0.307) in 16 trials; the respective proportions were similar for the alternative definition of season.

Male *N. squamipes* appeared to be more infectious than females, infecting a median 65% (IQR: 62.4, 65.7, n = 12) of blood-fed sand flies compared to 43% (IQR: 5.1, 59.9, n = 13) infected by females (z = -4.16, p<0.001). No differences were detected between age-classes (z<-0.74, p>0.456). Aggregated data for the rodent species other than *N. squamipes* indicated similar trends: a non-significant difference between age-classes (z<-1.02, p>0.308), and a tendency for males to be more infectious to sand flies than females: 52% (IQR: 44.7, 65.7, n = 3) *versus* 55% (IQR: 24.8, 63.4, n = 4) (z = -2.16, p = 0.031).

### Clinical *L. (V.) braziliensis* loads in sand flies and infectious rodents

The frequency distributions of the $\log_{10}$ transformed parasite loads in sand flies post xenodiagnosis were over-dispersed (S5C Fig). There was no correlation between the parasites loads in rodent blood and in the xenoinfected sand flies (Fig 4), nor in the contemporary paired samples for individuals ($r_s^2$ = 0.076, p = 0.602). Likewise, rodent blood parasite loads were not correlated with the proportion of exposed sand flies infected at point xenodiagnosis ($r_s^2$ = 0.234, p = 0.106). Few skin biopsy samples proved positive by qPCR for comparative analyses.

### Discussion

Previous field studies in the Zona da Mata in Pernambuco, Brazil, detected *Leishmania* infection in a mosaic of wild and synanthropic rodents and domesticated animals, indicating that they may play a role in zoonotic transmission of *L. (V.) braziliensis* [7]. The present CMR study in this region focussed on rodents as they represented the largest biomass [7,24]. *L. (V.) braziliensis* infections were confirmed in 603 individuals of six of the eight rodent species

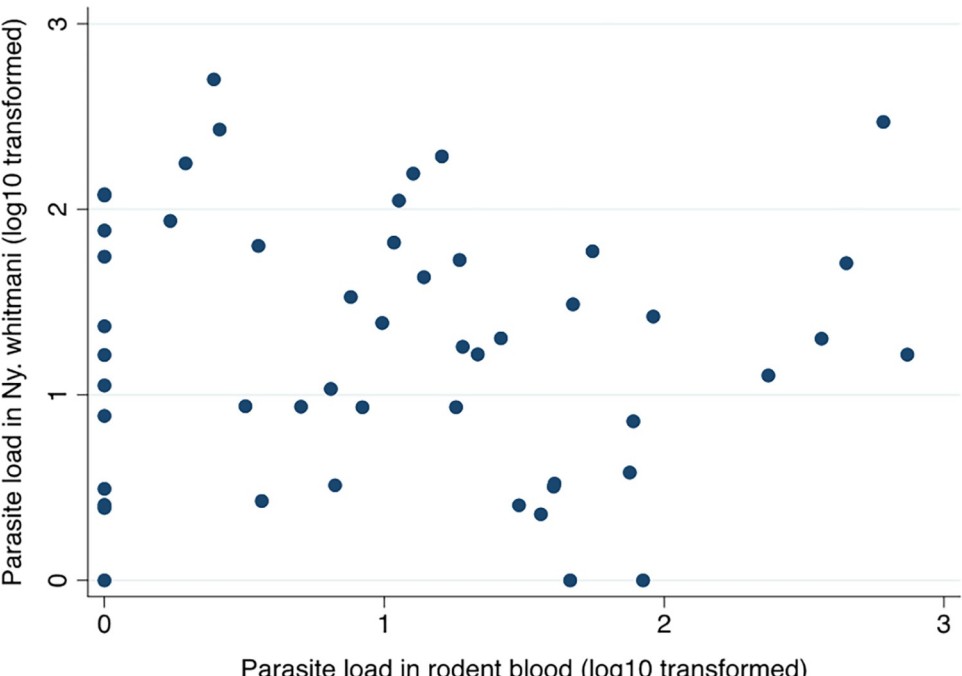

**Fig 4. The association between the $\log_{10}$ transformed *L. (V.) braziliensis* parasite loads estimated by qPCR in rodent blood samples and in *Ny. whitmani* sand flies infected during xenodiagnosis.**

captured. Period prevalence ranged from 12.5% to 61.5% being statistically higher in *N. squamipes* and *H. sciureus* populations (Table 2). No infections were detected in the two *Ol. nigripes* nor four *C. subflavus* individuals captured.

The epidemiological significance of rodent infections was further investigated by xenodiagnosis allowing the known vector *Ny. whitmani* to freely blood-feed on *N. squamipes, Ne. lasiurus, R. rattus, A. cursor* and *O. dasytrichus*. All individuals of the five species examined proved infectious except two individual *N. squamipes* in 3 xenodiagnosis experiments (Table 7). The median proportion of blood-fed flies infected by each species was relatively high (>45%) with exception of *O. dasytrichus* (1%), suggesting that at least the more abundant rodent species (*N. squamipes* 41% of total individuals captured, *R. rattus* 25%, and *Ne. lasiurus* 14%), significantly contribute to transmission. As the blood-fed *Ny. whitmani* from these experiments were killed and preserved for molecular testing immediately after blood-feeding, the results reflect potential infectiousness rather than transmissibility *per se* as the latter would require detection of metacyclic promastigotes post extrinsic incubation. Xenodiagnosis using the known permissive species, *Lu. longipalpis* [29], a possible secondary vector of *L. (V.) braziliensis* [10], demonstrated the infectiousness of all 3 *N. squamipes* and 2 *R. rattus* individuals upon dissection of flies 5–7 days after feeding. In this case, the results more likely reflect transmissibility.

Xenodiagnosis data for naturally infected wildlife hosts of *Leishmania* are few, and for *L. (V.) braziliensis* are limited to a single experimental infection study of *N. squamipes, Ne. lasiurus* and *R. rattus* using rodent colonies established with wild-caught animals from the current study region [12]. In that study, 10/18, 18/18 and 6/18 individuals respectively were infectious to *Lu. longipalpis*, results which help support the findings of the current study. The proportions of sand flies infected by these naturally and experimentally infected rodents were substantially higher than reported from xenodiagnosis experiments on regionally suspected rodent species experimentally infected with *L. major* strains, *L. donovani, L. tropica* or *L.(V.)*

*panamensis* [30–32], or lagomorphs and wild canids naturally infected with *L. infantum* [8,28,33].Some of the variation is attributed to the different detection techniques and timing; molecular methods are generally more sensitive than direct microscopic examination [7]. Notwithstanding, the proportion of flies infected here approach those reported for European and Latin American sand fly vectors infected with *L. infantum* by domestic dogs, the proven reservoir of *L. infantum* [8, 9].

Additional criteria to incriminate a reservoir host is that it can maintain infection through the non-transmission period i.e. during periods of low or zero vector abundance [9]. The recaptured rodents in this study showed chronic parasitic infections in the absence of any apparent clinical signs. The average duration of detected infection in individuals and in the whole rodent population (Fig 3), exceeded the period of seasonally low abundance of *Ny. whitmani*, which is variably 3–5 months from May to July/October in this region [19]. Targeting the *N. squamipes* population for repeat xenodiagnosis, we demonstrate persistence transmission potential over the 13-month observation period, infecting a median 48% (IQR: 30.1%-64.2%) of exposed *Ny. whitmani*, and observed significantly greater levels of infectiousness during the reported periods of low vector abundance compared to during the periods of high abundance (Fig 3). Since the rodent species occurred in sympatry, we were unable to demonstrate that *L. (V.) braziliensis* can be maintained by any single species, but comparative longitudinal xenodiagnoses studies would be useful to partition the basic case reproduction number $R_0$ to determine their relative epidemiological contribution to transmission [8].

In a related study and complimentary analyses of the current data, we demonstrate a high force of infection λ = 0.22/month (SD: 0.183) in the rodent population as a whole, with 50% of the population infected as juveniles (2–3 months old), and 80% infected by 8 months old, associated with a short average prepatent period (time to detectable infection) of 1–3 months [34]. The longitudinal infection incidence appeared highest in *Ne. lasiurus* and lowest in *R. rattus* but the variation was broad excluding detection of statistical differences between species [34].

Infected rodents were (re)captured in all the major habitat types under study and where humans frequent, suggesting that humans are also at risk of infection with *L. (V.) braziliensis*. The household plantations, typically comprising bananas, pineapples and vegetable cash crops, proved to be the habitat of greatest rodent abundance, infection prevalences, and relative infection risk (Tables 4 and 5). Vector presence in each of these ecological habitats is well documented [19]. We did not sample the commercial monoculture sugarcane plantations which are extensive in the region, as human presence is limited to seasonal planting and harvest periods by a seasonal labour force. Sand flies are absent in this habitat but on rare occasions rodents have been captured.

Crude infection prevalences were similar or possibly higher in Atlantic forest patches for some rodent species, notably *A. cursor*, *N. squamipes* and *R. rattus*. The two habitats yielded not dissimilar rodent species compositions, but importantly we show that 22.6% of individuals captured at least twice were captured in more than one habitat type including 19.8% of *N. squamipes* and 66.3% of *R. rattus*, indicative of the likely connectivity between these habitats (Table 6). Of these animals, 78.6% were infected at some point during the study. *R. rattus* linked the peri domestic environment to both plantation and forest. Thus, infected rodent dispersions between habitats occurs and we speculate that individual animal home ranges within the different habitats overlap that of sympatric species, thereby providing opportunities for inter-species and inter-habitat transmission. Many of the species in this study have been recorded in a diversity of habitats, with or without *L. (V.) braziliensis* infections, indicating their ecological adaptability [24,35–42].

It is also conceivable that there are concurrent domestic and peri domestic/sylvatic transmission cycles in this region each involving one (or more) primary host species. In this case,

the cycles would be linked by the vector *Ny. whitmani* which is regularly trapped in all three habitats [19]. The evidence to date suggests that *N. squamipes* and *R. rattus* may represent the primary hosts in these putative cycles respectively but this cannot be confirmed by the current data. Only *R. rattus* was captured in and around houses, and in all three sampled ecotypes under study; it is a synanthropic species frequently observed in the walls and ceilings of houses and has consistently shown high *Leishmania* infection rates in this focus [19] and elsewhere [24,43]. Additional domestic hosts may include dogs of which 20% were previously shown PCR positive [7] with higher prevalences reported elsewhere [43], though xenodiagnosis experiments have yet to be conducted. *N. squamipes* is physiologically adapted to a semi-aquatic life, though recognised as a generalist using aquatic and terrestrial environments alike [44]. It was the largest rodent species in this study (body weight range: >170–400 gm) and the most abundant species, which could be important for disproportionately attracting blood-seeking sand flies. Five species of *N. squamipes* occur over an extensive area of the Americas [45,46] suggesting that they may play a primary role in *L. (V.) braziliensis* transmission beyond northeast Brazil. Seasonally stable populations dominate the small mammal assemblage in eco-logically diverse habitats elsewhere [44,47,48], though not exclusively so; species compositions clearly vary between ecological foci [24,35,37,43] and possibly through time. For example, in a different ATL focus in Pernambuco, 52% of 46 small mammals captured in 2003 were marsu-pials (33.3% infected) compared to 0% of 118 animals captured in 2006 using the same trap-ping methods [24].

Xenodiagnosis is the gold standard to measure host infectiousness, but it is logistically diffi-cult to perform at scale. Quantification of *L. (V.) braziliensis* parasite loads in rodent blood by qPCR was found to be imprecise as a potential surrogate marker of host infectiousness. Indeed, there was generally poor correlation between corresponding parasitaemia levels in rodent blood and skin samples, and between skin samples and xenopositive sand flies. On the other hand, 83% of rodents with blood positive for *L (V.) braziliensis* were xenopositive suggesting that qPCR positivity may be a useful qualitative indicator of infectiousness.

In contrast to the surprisingly few ear tissue samples being qPCR positive, the *L. (V.) brazi-liensis* experimental study identified parasite loads in rodent ear tissue to be the best indicator of infectiousness to *Lu. longipalpis* [12]. Such an association has been shown also for cohorts of dogs naturally infected with *L. (L.) infantum* [49]. Despite reports of greater PCR sensitivity testing rodent blood compared to rodent skin [41,50], the current results may be due to unidentified technical reasons. We infer this from the success of molecular detection of *L. (V.) braziliensis* in ear skin similar to in spleen and liver samples from *N. squamipes*, *Ne. lasiurus* and *R. rattus* experimental animals [12], in naturally infected rodent populations [43], and in longitudinal follow-up of rodents experimentally infected with other *Leishmania* species, where the ear pinnae inoculation site remained a significant source of parasites [12,51,30]. Fur-thermore, finding significant *L. (V.) braziliensis* parasite loads in both skin and internal organs of experimental [12] and naturally infected [37,43] wild rodents in Brazil, and the high parasite loads in the blood of the current study animals, suggests that metastasis may be an intrinsic character of *L. (V.) braziliensis* in rodents, as is the case in humans and other animals [52–54]. Parasite tropism and their non-continuous distributions across the skin [55] will clearly influ-ence the accessibility to blood-feeding sand flies, and thus the hosts' infectious potential.

The chronic and asymptomatic nature of *Leishmania* infection in wild rodents observed in this population is not atypical [7,43,56,57]. Even following experimental inoculation of *N. squamipes*, *Ne. lasiurus* and *R. rattus* with high dose (5.5 x $10^6$/ml) *L. (V.) braziliensis*, the skin lesions that developed in 13/26 animals self-resolved within one month of onset [12]. In that study, all animals were infectious to *Lu. longipalpis* following spontaneous clinical recovery. Similarly, suspect rodent reservoir species experimentally infected with *L. major* or *L. donovani*

presented largely asymptomatic infections for 20–30 weeks pre-necropsy; all three *L. major* infected rodent species were infectious when asymptomatic [30,58].

## Conclusions

The collective data on infection prevalence, infectiousness and rodent spatial abundance support the view that the rodent assemblage in this focus represents an important wildlife source of *L. (V.) braziliensis* for onward transmission. *N. squamipes* at least plays a significant role, and probably also *R. rattus*, *Ne. lasiurus*, and *A. cursor*. Peri domestic plantations appear to be locations of highest infection risk for rodents, and by extrapolation, humans, forming one part of a wider eco-epidemiological network linking domestic and forest habitats. *L. (V.) braziliensis* infections are documented in a multitude of other wild animal species in this region and elsewhere (reviewed [59]) but infectiousness studies remain to be conducted. A complete understanding of the geographical complexity of enzootic transmission needs to evaluate the relevance of reported mixed *Leishmania* spp. host infections [24, 43] and regional genetic diversity of *L. (V.) braziliensis* [60]. In Pernambuco three genetic groups have so far been identified [61], one and two, that have high heterozygosity, occur in areas where *Ny. whitmani* is the vector, such as Amaraji. The third variant is associated with forests where other sand fly species are suspected vectors. Transmission pathways ultimately depend on the prevailing conditions of the ecosystem and climatic drivers, undoubtedly fuelled by environmental modifications by humans.

## Supporting information

**S1 Fig. Satellite image showing the location of the capture areas Engenho Raiz de Dentro (8° 42.7944'S, 35° 44.7078'W) and Engenho Refrigério (8° 42.7963'S, 35° 49.4742'W) in relation to Amaraji town, Amaraji municipality, Pernambuco State, Brazil.**
(DOCX)

**S2 Fig. A homestead in the Engenho Raiz de Dentro area. The open garage on the left side of the house was used as a field laboratory.**
(DOCX)

**S3 Fig. A homestead in the Engenho Refrigério area. Note the proximity of the banana plantation to the house and on the other side of the lane a fenced field for livestock.**
(DOCX)

**S4 Fig. Frequencies of recaptures of rodent species over the 20 trapping sessions. (A)** *Ne. lasiurus*, *N. squamipes* **and** *R. rattus***; (B)** *A. cursor*, *H. sciureus* **and** *O. dasytrichus***; (C)** *N. squamipes* **in forest and plantations; (D)** *R. rattus* **in plantations, forest, and the domestic environment.**
(DOCX)

**S5 Fig. Frequency distributions of** *L. (V.) braziliensis* **parasite loads detected by qPCR in (A) rodent blood (per 200µL); (B) ear skin biopsies (per 25mg); (C)** *Ny. whitmani* **sand flies (per individual fly). In (A)** *N. squamipes* **alone (right), all other rodent species (left). Note the differences in the Y-axis scales.**
(DOCX)

**S6 Fig. Boxplot of the median (IQR)** *L. (V.) braziliensis* **parasite loads detected by qPCR in rodent blood samples (dark bars) and** *Ny. whitmani* **sand flies post exposure to rodents during xenodiagnosis (light bars).**
(DOCX)

**S1 Table. Trapping schedule showing the number of trap-nights (total single traps set over trapping period).**
(DOCX)

**S2 Table. Median recapture interval (days) and number of times individual rodents were recaptured.**
(DOCX)

**S3 Table. Infection prevalences based on conventional PCR compared to quantitative PCR (qPCR) of rodent blood samples using PCR target primers specific to parasites of the genus *Leishmania* (*Viannia*) *braziliensis*.**
(DOCX)

**S4 Table. Infection and infectiousness of individual rodents at the time of xenodiagnosis exposing *Ny. whitmani* sand flies, and the associated *L.* (*V.*) *braziliensis* parasite loads in the rodent blood and xenopositive sand flies.**
(DOCX)

**S5 Table. Infection and infectiousness of individual rodents at the time of xenodiagnosis exposing *Lu. longipalpis* sand flies, and the associated *L.* (*V.*) *braziliensis* parasite loads in the rodent blood and xenopositive sand flies.**
(DOCX)

## Acknowledgments

We are grateful to biology students Mariana Verissimo, Priscila Carla da Silva, Luana Patrícia Brito for their laboratory support. Hélio Valença, Amilton Silva, (technicians of FUNASA/MS —National Health Foundation/Ministry of Health) and Fernando Silva (technician of IAM/ FIOCRUZ) for their support in the field work. Also to Maria Edileuza Felinto Brito, Éricka Almeida and Andréa Sales (IAM/FIOCRUZ) for their support in the laboratory, and Romário Melo and José Egberto Silva for supporting work with rodents and sandflies. We would also like to thank Professor James L. Patton (Museum of Vertebrate Zoology, University of California, Berkeley, USA) for his help with the rodent taxonomy. We are grateful also for institutional and logistical support from IAM/Fiocruz to vehicle availability for field work.

## Author Contributions

**Conceptualization:** Sinval Pinto Brandão-Filho.

**Data curation:** José Ferreira Marinho-Júnior.

**Formal analysis:** Jeffrey Shaw, Orin Courtenay.

**Funding acquisition:** Sinval Pinto Brandão-Filho.

**Investigation:** José Ferreira Marinho-Júnior, Juliana F. C. L. S. Monteiro, Francisco Gomes de Carvalho, Jeffrey Shaw, Orin Courtenay.

**Methodology:** José Ferreira Marinho-Júnior, Juliana F. C. L. S. Monteiro, Ana Waléria Sales de Carvalho, Francisco Gomes de Carvalho, Milena de Paiva Cavalcanti, Jeffrey Shaw, Orin Courtenay.

**Supervision:** Sinval Pinto Brandão-Filho.

**Validation:** Orin Courtenay.

**Writing – original draft:** José Ferreira Marinho-Júnior, Sinval Pinto Brandão-Filho.

**Writing – review & editing:** Jeffrey Shaw, Orin Courtenay, Sinval Pinto Brandão-Filho.

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
