## [Decision Letter · Decision Letter 0]

23 Jun 2022

Dear Dr. Brandão Filho,

Thank you very much for submitting your manuscript "Chronic asymptomatic infection associated with high levels of infectiousness of wild rodent communities highlights their role in the epizootiology of human American Tegumentary Leishmaniasis: a capture-mark-recapture study in Brazil" for consideration at PLOS Neglected Tropical Diseases. As with all papers reviewed by the journal, your manuscript was reviewed by members of the editorial board and by several independent reviewers. In light of the reviews (below this email), we would like to invite the resubmission of a significantly-revised version that takes into account the reviewers' comments. 

The manuscript by Marinho-Junior et al presents a comprehensive reservoir collection capture that provide important information for the specialized Leishmania/sand fly community. Presentation and discussion of the data should be optimized in order to favor the manuscript: 

1) Points raised by the referees should be answered carefully. 

2) The data comparison blood vs. skin qPCR is not helping the discussion and presentation/interpretation of the most important information present in the manuscript: the reservoirs. I suggest the authors combine the positivity of results to create a cut-off of positivity calling radar than compare the blood vs biopsy values. The skin location my not be adequate for parasite search and its extension as well not. This particular data comparison detract from the main point of the paper. Due that, Fig 4 is not helping and can be combined with S6 if authors want to keep.

3) Infectiousness to sand flies should be better explained. Why the significant variation in the number of flies used for the xenodiagnostic? 

As suggestion to improve the manuscript readability, the authors should consider:

4) A shorter title. 

5) (line32) "infectiousness to" to "infectiousness potential to". It is more accurate to the data.

6) The biology of sand flies multiple blood meals could help the manuscript interpretation. It's a pivotal biology fact of this insect vector that should not be neglected on this manuscript discussion.

7) lines 118-122. This entire paragraph is not accurate to literature available. Is highly speculative and not helping the manuscript main point.

8) Conclusions are important and adequate. The discussion should be simplified to match that.

We cannot make any decision about publication until we have seen the revised manuscript and your response to the reviewers' comments. Your revised manuscript is also likely to be sent to reviewers for further evaluation.

Sincerely,

Tiago Donatelli Serafim

Associate Editor

Jesus Valenzuela

Deputy Editor

The manuscript by Marinho-Junior et al presents a comprehensive reservoir collection capture that provide important information for the specialized Leishmania/sand fly community. Presentation and discussion of the data should be optimized in order to favor the manuscript: 

1) Points raised by the referees should be answered carefully. 

2) The data comparison blood vs. skin qPCR is not helping the discussion and presentation interpretation of the most important information present in the manuscript: the reservoirs. I suggest the authors combine the positivity of results to create a cut-off of positivity calling radar than compare the blood vs biopsy values. The skin location my not be adequate for parasite search and its extension as well not. This particular data comparison detract from the main point of the paper. Due that, Fig 4 is not helping and can be combined with S6 if authors want to keep.

3) Infectiousness to sand flies should be better explained. Why the significant variation in the number of flies used for the xenodiagnostic? 

As suggestion to improve the manuscript readability, the authors should consider:

4) A shorter title. 

5) (line32) "infectiousness to" to "infectiousness potential to". It is more accurate to the data.

6) The biology of sand flies multiple blood meals could help the manuscript interpretation. It's a pivotal biology fact of this insect vector that should not be neglected on this manuscript discussion.

7) lines 118-122. This entire paragraph is not accurate to literature available. Is highly speculative and not helping the manuscript main point.

8) Conclusions are important and adequate. The discussion should be simplified to match that.

Reviewer's Responses to Questions

**Key Review Criteria Required for Acceptance?**

**Methods**

-Are the objectives of the study clearly articulated with a clear testable hypothesis stated?

-Is the study design appropriate to address the stated objectives?

-Is the population clearly described and appropriate for the hypothesis being tested?

-Is the sample size sufficient to ensure adequate power to address the hypothesis being tested?

-Were correct statistical analysis used to support conclusions?

-Are there concerns about ethical or regulatory requirements being met?

Reviewer #1: The authors have met all method criteria to have the manuscript accepted.

Reviewer #2: It is not clear how the exposure of the captured rodents to wild infected sand flies impact in posterior prevalence when they are recaptured. Unless it is a misunderstanding, I think this is a serious problem in the design of this manuscript. It is also not clear how this was considered in terms of ethical use of research animals. Besides that, this problem is specially important if we consider the recent findings of Valenzuela´s group about the change in Leishmania development and differentiation after a second blood meal. It is impossible to know the proportion of sand flies that might have been considered not-infected in the control screening but appear as infected in the xenodiagnosis due to amplification of parasite populations.

Reviewer #3: Most of the study is well designed, and the methods are clearly and thoroughly described. The sample of 603 rodents, captured/recaptured in 20 trapping rounds over two years (9.920 single trap nights) is sufficient to show rodent species diversity and infection prevalence in three study habitats. Statistical analyses are well performed and well described. 

The only problematic method is the use of wild-caught Ny. whitmani females for xenodiagnosis. To calculate infection rates and parasite loads in xenodiagnoses, authors subtracted the corresponding values for wild-caught flies from the experimental data. Therefore, the resulting data may be affected by unequal representation of naturally infected flies among sand fly groups applied to individual animals. Also, parasite loads in naturally infected flies with mature infection would be much higher than in freshly engorged females during the experiment. Fortunately, the authors also used L. longipalpis flies from the colony in 5 experiments and these experiments did not yield substantially different results (Table 7, S4 and S5). Above mentioned facts should be emphasized in the Discussion and the Abstract: xenodiagnoses results gained by wild-caught sand flies should be treated with extreme caution and exclusively sand flies from the colony should be used for future studies.

**Results**

-Does the analysis presented match the analysis plan?

-Are the results clearly and completely presented?

-Are the figures (Tables, Images) of sufficient quality for clarity?

Reviewer #1: Yes, the analysis presented match the analysis plan and the results are clearly and completely presented. The tables and figures are of sufficient quality; however, photos and Google map image of the region are redundant and should be excluded, since methods describe in detail the areas of study.

Reviewer #2: The analysis is straightforward, the results and figures are well presented.

Reviewer #3: Results are clearly presented, and all the figures and tables are of sufficient quality (minor suggestions are in the section Editorial Modifications).

**Conclusions**

-Are the conclusions supported by the data presented?

-Are the limitations of analysis clearly described?

-Do the authors discuss how these data can be helpful to advance our understanding of the topic under study?

-Is public health relevance addressed?

Reviewer #1: Yes, the conclusions are supported by the data presented, limitation of analysis particularly Xenodiagnoses using field collected Ny. whitmani were justified and the authors discuss how their findings can be helpful to advance our understanding of the topic under study. The manuscript has an important public health relevance; however, the authors should have clearly stated in the discussion how their findings can help public health authorities in Brazil to prevent the spread of Leishmania (Viannia) braziliensis in the endemic area.

Reviewer #2: The results are relevant in a very broad and qualitative sense. However, the impact of the exposure to wild infected sand flies in my opinion clearly impacts the reliability of the quantitative results.

Reviewer #3: See the Summary and General Comments section.

**Editorial and Data Presentation Modifications?**

Reviewer #1: Minor modifications are needed in order to address typo mistake and presence of figures irrelevant for the understanding of the manuscript.

Reviewer #2: (No Response)

Reviewer #3: Table 1, 6 and S1: “Peridomestic” or “domestic” should be replaced with “Peri domestic” 

Figure S2 can be omitted because it does not provide more information than Figure S3 (however, since it is included in the supplementary material, redundancy may not be an issue).

Line 93: Are authors sure these two non-English literary sources should be cited here (2, 3)?

Line 114: The mentioned few experimental studies like de Moura et al (2005, doi.org/10.1128/IAI.73.9.5827-5834.2005) should be cited.

Table 1: Ol. nigripes

Lines 258-267: Negative control should be specified for qPCR.

Line 332: According to Fig S4b, H. sciureus and O. dasytrichus have been recaptured up to 5 times.

Table 3: The blood volume 0.2 mL (not 0.2µL)

Discussion on lines 541-546: The study “Christensen and Herrer 1972 (TRANS R SOC TROP MED HYG 66(5))” showing the infectiousness of Choloepus hoffmanni infected with L braziliensis to vectors should also be cited.

Discussion on lines 631-638: The authors should add to the discussion that another reason for the higher positivity values of the tissue samples from the ears in the laboratory study may be that in this field study only half the tissue sample was taken (25 vs 50 mg)

**Summary and General Comments**

Reviewer #1: Dear editor of Plos. Neglected Tropical Diseases,

I would like to thank you very much for the opportunity to review the manuscript of Dr. Marinho-Junior et al. PNTD-D-22-00244- “Chronic asymptomatic infection associated with high levels of infectiousness of wild rodent communities highlights their role in the epizootiology of human American Tegumentary Leishmaniasis: a capture-mark-recapture study in Brazil.

Apart from a small typo mistake in line 48, the paper is very well written and, in my view, meets the proper usage of English language standards of Plos, Neglected Tropical Diseases, granting it merits for publication. 

Briefly, the authors studied the epidemiological significance of wildlife infections among guild of rodent reservoirs present in three distinct ecotopes in two municipalities in the city of Amaraji, State of Pernambuco, Brazil, for the presence of Leishmania (Viannia) braziliensis by employing capture-remake-recapture (CMR), traditional PCR, qPCR and xenodiagnosis methods aiming to address five different goals. The results show a highly interesting and unique overview of the dispersion of infected putative rodent reservoirs among the three distinct ecotopes with the participation of the natural vector Ny. whitmani. The data presented can assist local public health authorities in determining the correct strategies for reducing the transmission of L.(V) brazilinesis in the region, which confirms the high relevance and scientific contribution of the manuscript. 

According to my interpretation of the discussion the authors seem to believe in independent transmission cycles occurring in each ecotope, considering the free movement of reservoirs, hosts and vector(s) in the region, which might help understand what happens to parasite and vector(s) during the non (low) and high transmission periods. This alone is a novel approach to the traditional eco-epidemiology of Leishmaniasis and I applaud the authors for it. 

A few comments that I would like to make, without diminishing the extremely laborious and time-consuming field and laboratory work of the authors, are:

1- The authors should have included a table containing the sand fly fauna present in each ecotope, although not in the five main goals of the paper the distribution, frequency and seasonality of sand fly species per ecotope would have enriched the manuscript. 

2- In addition, it would have been extremely valuable if the authors had collected the most abundant sand fly species of each ecotope (probably Ny. whitmani) and scrutinized their blood meal content before using some sand flies for xenodiagnoses. It is well known now that a second and successive blood meals augment the parasite infectivity in the sand fly, at least in laboratory conditions, please refer to the work of Serafim et al, Nat Microbiol. 2018. Also, the authors might be able to evaluate the reservoir preference of Ny.whitmani with the three most abundant rodent species present in the three distinct ecotopes, if blood meals were analyzed. 

2- Another highly desired approach, would have been for the authors to bring Ny, whitman to the lab and rear the species for one generation, since it seems to be extremely difficult to establish a Ny. whitmani colony in captivity, before taking some flies back to the filed for feeding on the three most abundant rodent reservoir species. This, relatively simple method, would have shed some light on the role of Ny. whitmani as the main vector in the area.. 

3- The 6% naturally infected sand flies observed during standardization of xenodiagnoses to count for potential Leishmania background level, since authors initially used field collected Ny. whitmani, seems to be extremely high considering other field studies in Brazil. Did the authors collect the flies from the ecotope where most of infected reservoirs were present (plantations) or authors just did a pool containing flies from all three ecotypes? Please, clarify it in the methods. 

4- Why did the authors not attempt to transmit L(V) braziliensis to hamsters in the lab by the bite of Ny. whitmani used for xenodiagnoses? Did the authors try to maintain Ny. whitmani in the lab after xenodiagnoses for observing in vivo parasite differentiation and development? It would have demonstrated the degree of infectivity and transmissibility of this vector, satisfying the criteria proposed by Killick-Kendrick to incriminate a sand fly as a vector. 

5- I assume that during editing of the paper lines 343, 380,387, 431,465 and 513 will be deleted. 

6- I don’t think it is necessary to include the Google map location and photos of the area, since the localities are very well described in the methods. 

Sincerely,

Referee A

Reviewer #3: The manuscript elucidated the reservoir system of L. braziliensis in Pernambuco, NE Brazil. The authors studied rodent communities in three interconnected habitats and described the involvement of each rodent species in the transmission of the parasite, with a precise description of their habitat association and infection risk in different habitats. Long-term maintenance of infections was documented by recapture of infected animals and their persistent infectiousness to vectors using xenodiagnosis. The manuscript is very well written; it is based on a sufficient amount of material that is perfectly statistically evaluated and presented. I suggest minor revision of the manuscript (see my comment about xenodiagnoses with wild-caught sand flies).

PLOS authors have the option to publish the peer review history of their article (what does this mean?). If published, this will include your full peer review and any attached files.

Reviewer #1: No

Reviewer #2: No

Reviewer #3: No
---

## [Decision Letter · Decision Letter 1]

29 Nov 2022

Dear Dr. Brandão-Filho,

We are pleased to inform you that your manuscript 'High levels of infectiousness of asymptomatic Leishmania (Viannia) braziliensis infections in wild rodents highlights their importance in the epidemiology of American Tegumentary Leishmaniasis in Brazil.' has been provisionally accepted for publication in PLOS Neglected Tropical Diseases.

Best regards,

Tiago Donatelli Serafim

Academic Editor

Jesus Valenzuela

Section Editor

<style type="text/css">p.p1 {margin: 0.0px 0.0px 0.0px 0.0px; line-height: 16.0px; font: 14.0px Arial; color: #323333; -webkit-text-stroke: #323333}span.s1 {font-kerning: none

</style>

Reviewer's Responses to Questions

**Key Review Criteria Required for Acceptance?**

**Methods**

-Are the objectives of the study clearly articulated with a clear testable hypothesis stated?

-Is the study design appropriate to address the stated objectives?

-Is the population clearly described and appropriate for the hypothesis being tested?

-Is the sample size sufficient to ensure adequate power to address the hypothesis being tested?

-Were correct statistical analysis used to support conclusions?

-Are there concerns about ethical or regulatory requirements being met?

Reviewer #1: (No Response)

Reviewer #2: (No Response)

Reviewer #3: (No Response)

**Results**

-Does the analysis presented match the analysis plan?

-Are the results clearly and completely presented?

-Are the figures (Tables, Images) of sufficient quality for clarity?

Reviewer #1: (No Response)

Reviewer #2: (No Response)

Reviewer #3: (No Response)

**Conclusions**

-Are the conclusions supported by the data presented?

-Are the limitations of analysis clearly described?

-Do the authors discuss how these data can be helpful to advance our understanding of the topic under study?

-Is public health relevance addressed?

Reviewer #1: (No Response)

Reviewer #2: (No Response)

Reviewer #3: (No Response)

**Editorial and Data Presentation Modifications?**

Reviewer #1: (No Response)

Reviewer #2: (No Response)

Reviewer #3: (No Response)

**Summary and General Comments**

Reviewer #1: (No Response)

Reviewer #2: (No Response)

Reviewer #3: All my minor comments have been incorporated in the new version of the text. I gladly agree to publish the manuscript.

PLOS authors have the option to publish the peer review history of their article (what does this mean?). If published, this will include your full peer review and any attached files.

Reviewer #1: No

Reviewer #2: No

Reviewer #3: No

---

## [Editor Report · Acceptance letter]

31 Dec 2022

Dear Dr. Brandão-Filho,

We are delighted to inform you that your manuscript, "High levels of infectiousness of asymptomatic Leishmania (Viannia) braziliensis infections in wild rodents highlights their importance in the epidemiology of American Tegumentary Leishmaniasis in Brazil.," has been formally accepted for publication in PLOS Neglected Tropical Diseases.

Best regards,

Shaden Kamhawi

co-Editor-in-Chief

Paul Brindley

co-Editor-in-Chief
